# Mandibular Titanium Miniplates Change the Biomechanical Behaviour of the Mandible in the Case of Facial Trauma: A Three-Dimensional Finite Element Analysis

**DOI:** 10.3390/bioengineering10090994

**Published:** 2023-08-22

**Authors:** Nicolas Graillon, Jean-Marc Foletti, Yves Godio-Raboutet, Laurent Guyot, Andrea Varazzani, Lionel Thollon

**Affiliations:** 1Laboratoire de Bioméchanique Appliquée (LBA), Gustave Eiffel University/Aix-Marseille University, 13015 Marseille, France; jeanmarc.foletti@ap-hm.fr (J.-M.F.); yves.godio-raboutet@univ-eiffel.fr (Y.G.-R.); laurent.guyot@ap-hm.fr (L.G.); lionel.thollon@univ-eiffel.fr (L.T.); 2Department of Oral and Maxillofacial Surgery/Assistance Publique-Hopitaux de Marseille (APHM), Conception University Hospital, 13005 Marseille, France; 3Maxillo-Facial Surgery, Facial Plastic Surgery, Stomatology and Oral Surgery, Hospices Civils de Lyon, Lyon-Sud Hospital—Claude-Bernard Lyon 1 University, 69310 Pierre-Benite, France; andrea.varazzani@chu-lyon.fr

**Keywords:** mandible fracture, facial trauma, internal fixation, mandibular osteosynthesis, implant removal, finite element analysis

## Abstract

Our study aimed to compare the biomechanical behaviour of mandibles with or without titanium miniplates when subjected to an impact after bone healing using a finite element model (FEM) of the human mandible. We simulated mandibular trauma on an FEM of a human mandible carrying or not two parasymphyseal miniplates and applying a concentrated force of 2000 N to four different areas, including the insertion area, the area straddling the edge of the miniplates and the adjacent bone, at a distance from the miniplates on the symphysis, and on the basilar border of the mandible below the miniplates. Then, we compared the Von Mises stress distributions between the two models. In the case of an impact on the miniplates, the maximum Von Mises stress occurred in two specific areas, on the cortical bone at the posterior border of the two miniplates at a distance from the impact, while in the model without miniplates, the Von Mises stresses were homogenously distributed in the impact area. The presence of titanium miniplates in the case of trauma affects the biomechanical behaviour of the mandible and could cause more complex fractures. We recommend informing patients of this potential risk.

## 1. Introduction

The use of titanium miniplates is the gold standard in the treatment of mandibular fractures and mandibular osteotomies [1]. Nevertheless, systematic removal of asymptomatic miniplates after bone healing remains controversial. On the one hand, titanium miniplates are biocompatible and therefore could be retained without biological consequences, thereby avoiding a second surgery, its potential complications, and inherent additional costs [2,3,4]. On the other hand, the biomechanical behaviour of the titanium miniplates in the case of trauma has not been evaluated. These miniplates could cause more complex fractures, depending on their location, or in relation to the risk of comminuted fracture or the presence of osseointegrated former miniplates [5,6]. The risk of fracture occurring on a mandible with miniplates has already been described [6,7,8], and is associated with more postoperative complications such as malocclusion, postoperative infection, osteomyelitis, and malunion [7]. This risk should not be neglected since mandibular fracture is one of the most common trauma injuries [9] and maxillofacial trauma recurrence has been widely described, mostly in patients with a history of interpersonal violence and alcohol and drug addiction [10,11,12,13]. Young patients having undergone orthognathic surgery and patients practicing contact or fighting sports or serving in the army are also more vulnerable to facial trauma [14,15,16]. In a previous experimental preliminary study [17], we showed that miniplates tend to reinforce the area located below them, transmitting forces to the vulnerable areas located at the edges of the miniplates and thereby causing complex fractures in the areas located close to the miniplates. These conclusions are in accordance with our clinical observations [6].

Numerical modelling may help us understand the stress distribution on the bone tissue and the bone implant surface while enabling us to simulate different types of trauma and osteosynthesis [18], in contrast to clinical studies [6] in which impact parameters could not be controlled. In addition, experimental studies, though enabling us to control the impact parameters, allowed few variations in variable parameters regarding trauma and miniplate location, which may be accounted for by the shortage of post mortem human subjects.

Furthermore, using finite element analysis (FEA), which is recognized as a valid and non-invasive method of predicting the biomechanical behaviour of the mandible in terms of stress, strain, and deformation [19] under physiological loadings or trauma, may provide further arguments in the debate over miniplate removal.

This study aimed at comparing the biomechanical behaviour of a mandible (with or without miniplates) subjected to an impact, according to the Von Mises stress distribution, using an FEA of a human mandible.

## 2. Materials and Methods

### 2.1. Model Description

We used a 3D finite element model (FEM) which had been previously developed and validated by our team [20]. DICOM data extracted from a CT scan of the head were computed using MIMICS software v12.3^®^ (Materialise, Louvain, Belgium) in order to recreate the skull’s geometry. The meshing was created using Hypermesh^®^ software (Altair engineering Inc., Detroit, MI, USA). The component properties were assigned using Hypercrash^®^ (Altair engineering Inc., Detroit, MI, USA), and the simulations were conducted using Radioss^®^ (Altair engineering Inc., Detroit, MI, USA). The skull base, facial bones’ geometry, and mesh patterns, including the glenoid fossae of both temporomandibular joints and the maxillary teeth and their alveolar surfaces, were reused from our initial model. We also retained the mandible’s geometry, but the meshing was refined thanks to a specific 3 mm thick cortical tetrahedral 3D meshing and a 3D tetrahedral meshing for the mandibular cancellous bone. Since the structure is complex, it was not possible to apply hexamesh; therefore, the tetrahedral mesh was used with an element of 1 mm. The meshes of the mandibular teeth and their alveolar surfaces were adapted. To ensure that the size of the mesh elements had no impact on the stress distribution, we compared the mesh used with 1 mm elements with a mesh using 0.5 mm elements.

Two 4-hole 1.0 mm thick 2.0 miniplates (Global D, Brignais, France) were placed on the right parasymphysis, simulating the previous treatment of a vertical simple parasymphysis fracture. We considered that the patient had a healed fracture of the right parasymphysis of the mandible, corresponding to a fully consolidated bone after callus remodelling following fracture that resulted in “restitutio ad integrum”. The geometry of the miniplates was extracted from the STL files furnished by their manufacturer and meshed using Hypermesh^®^ (Altair engineering Inc., Detroit, MI, USA). Five millimetre screws were designed with the same software. The meshing was continued with nodes placed between the miniplates, screws, and the maxillary bone and distributed in such a way so as to simulate the osseointegration of the implants.

The model was previously validated [20] according to experimental trials conducted by Schneider et al. [21] and Viano et al. [22] and by an experimental study in the Tuchtan et al. [20] study itself.

### 2.2. Contact Conditions

The mesh was continued between the teeth and bone, cancellous and cortical bone, cortical bone and miniplate, and cancellous bone and screws. Penalty contacts were applied between mandibular and maxillary teeth and between the mandible and facial bone to avoid penetration of one part by another.

### 2.3. Boundary Conditions

The skull base, including the glenoid fossae, the maxillary bone, and the maxillary teeth, was restrained in all directions. Condyle translation was blocked in all three directions and rotation was restrained except in the sagittal plane in order to facilitate the creation of FEM simulations of mouth opening and closure (Figure 1).

Jaw elevator muscle forces were simulated by creating a concentrated force applied to the mandibular insertions of the masseter (388.5 N), the temporal (333 N), and medial pterygoid muscles (432 N) using force vectors following the axis of contraction of each muscle in order to simulate muscle contraction as described in the literature [23].

### 2.4. Material Data

Our material data on the compact and cancellous bones, teeth, and titanium were extracted from the analyses conducted in previous studies [14,16,24,25,26,27,28,29,30,31,32,33,34]. These data are summarized in Table 1.

### 2.5. Trauma Simulation

Blunt trauma numerical simulations on modelled mandibles with and without miniplates were run on Altair Radioss (Altair engineering Inc., Detroit, MI, USA) using a concentrated force of 2000 N, which has been frequently used in the literature [25,26,35,36,37] and corresponds to the impact forces in boxing head impacts [22]. For the purposes of a fair comparison, for each simulation the concentrated forces were applied to the same nodes in each model (with or without miniplates), except for the nodes covered by miniplates and screws. In this case, the forces were applied to the surface of the miniplates and screw heads in the models with miniplates and on the bone in the models without miniplates. Four impact areas were tested (Figure 2): in the miniplate area, in the aera straddling the edge of the plates and the adjacent bone, at a distance from the miniplates on the symphysis, and on the basilar border below the miniplates. In each case, we compared the Von Mises stress distributions with the same impact on the mandible models with or without miniplates.

## 3. Results

The FEA results shown in our figures represent the color-coded Von Mises stress (MPa) distribution in the mandible with and without miniplates subjected to the corresponding impact loading.

The Von Mises stress distribution was relatively consistent between the tetrahedral mesh with 1 mm elements and the tetrahedral mesh with 0.5 mm elements (Figure 3). In particular, the stress distribution at the edges of the miniplate was similar.

Figure 4 shows the Von Mises stress distribution (MPa) in a direct impact on the miniplates compared to the same impact on the modelled mandible without miniplates. In this case, the Von Mises stress distribution demonstrated that the maximal stress area was superimposed upon the impact area (Figure 4A). In the model with miniplates, the Von Mises stress distribution showed that the two maximal stress areas were located on the cortical bone at a distance from the impact and on the posterior border of the two miniplates, with a 50% increase in the maximum Von Mises stress as compared to the same nodes on the model without miniplates. A maximum Von Mises value of 48 MPa was found on the nodes posterior to the alveolar miniplate versus 24 MPa on the model without miniplates. A maximum Von Mises value of 48 to 68 MPa was found on the nodes posterior to the basilar miniplate versus 30 MPa on the mandible without miniplates. The maximal Von Mises stress values were lower on the bone impact area, particularly under the miniplates. They were reduced by 50% under the alveolar miniplate (18 MPa under the alveolar miniplate versus 42 MPa on the model without miniplates) and by 60% under the basilar miniplate (18 MPa under the alveolar miniplate versus 48 MPa on the model without miniplates) (Figure 4B). On the impact area, the stresses were mainly applied to the miniplates (Figure 4C).

Figure 5 illustrates the Von Mises stress distribution (MPa) in an impact on the area straddling the posterior edge of the miniplates and the adjacent bone compared to the same impact on a mandible without miniplates. In the model without miniplates, the Von Mises stress distribution showed that the maximal stress area was superimposed upon the impact area (Figure 5A). In the model with miniplates, the Von Mises stress distribution showed two maximal stress areas located on the cortical bone at the posterior border of the two miniplates, with an increase in the maximum Von Mises stress of 20% compared to the model without miniplates (87 MPa in the model with miniplates versus 70 MPa in the model without miniplates). The maximal Von Mises stresses were higher in the model with miniplates (Figure 5B). By contrast, they were reduced by 30% on the bone under the miniplates (20 MPa on the bone under the miniplate versus 50 to 60 MPa on the model without miniplates). In the impact area, the Von Mises stresses were mainly applied to the miniplates (Von Mises maximum stress value 219 MPa) (Figure 5C).

Figure 6 illustrates the Von Mises stress distribution (MPa) in an impact on the symphysis at a distance from the miniplates compared to the same impact on the model without miniplates. The Von Mises stress distributions were relatively similar in the two models, except that the maximal Von Mises stresses were reduced by 30% on the bone around the screws under the miniplates (Figure 6B) and mainly applied to the miniplates and part of the screws located in the cortical bone (Figure 6C).

Figure 7 shows the Von Mises stress distribution (MPa) in an impact on the basilar border below the miniplates compared to the same impact on the model without miniplates. In this model, the Von Mises stress distribution showed a single maximal stress area located halfway up the mandible (Figure 7A). In the model with miniplates, the Von Mises stress distribution showed several maximal stress areas located on the cortical bone, at the posterior border of the basilar miniplate, and between the two miniplates (Figure 7B). The maximal Von Mises stress was reduced by 40% on the bone under the miniplates (Figure 5B). In the impact area, the maximal Von Mises stress was mainly applied to the basilar miniplate and the cortical part of the screws (Figure 7C).

Range stress histograms (Figure 8) showed that stress-level distribution was rather similar between the models without and with miniplates. Only in the case of an impact on the symphysis at a distance from the miniplates did the range stress histogram show a shift towards low stress values in the presence of the miniplates.

## 4. Discussion

Our study stands out as the first FEA study evaluating the biomechanical behaviour of osseointegrated mandibular miniplates in the case of trauma occurring after bone healing. We have established that miniplates changed the biomechanical behaviour of the mandible in our numerical model in the case of trauma involving the miniplate area.

Our finite element analysis showed that in the case of an impact in the miniplate area, the stresses were mostly absorbed by the miniplates, which tended to protect the underlying bone. However, these stresses were transmitted through the miniplates to the bone close to the edges of the miniplates, creating several areas of high-level stresses. These areas are exposed to an increased risk of fracture, which could result in complex fracture lines and in multi-fragmentary fractures located at the edge of the miniplates. From a biomechanical perspective, titanium is more rigid than bone, thereby allowing the miniplates to absorb stresses instead of the underlying bone. However, this rigidity does not allow dispersing stresses, which are transmitted through the miniplates to the bone on their edges, creating high levels of stress areas on this bone. In fact, in the case of an impact at a distance from the miniplates, stresses are absorbed by the miniplate, more rigid, and weakly transmitted to the bone. This could explain the shift towards low stress values in the stress range histograms of the bone in the case of an impact at a distance from the miniplates. In the case of an impact on the miniplates, the stress histogram showed no difference in term of stress-level distribution. However, the Von Mises distribution was different depending on the presence of miniplates, with less stresses on the bone under the miniplates and more stresses on the bone at the border at the miniplates. In this case of a direct impact on the miniplates, stresses were transmitted to the bone close to the edges of the miniplates.

Our findings are in accordance with our clinical observations of fractures occurring at the edges of miniplates on mandibles bearing miniplates (Figure 9) [6].

In this multicentre case series, we identified 13 patients presenting a fracture on a mandible bearing miniplates for at least 6 months. The indication of the former osteosynthesis was a fracture due to assault in nine cases, a horse hoof kick in one case, and a bilateral sagittal split osteotomy for the correction of dental class II in three cases. The mechanism of the second trauma was assault in 10 cases, including the 9 cases who initially presented with a fracture due to an assault, a subsequent hoof kick for the patient who had been kicked initially, and a traffic accident for 2 patients treated with bilateral sagittal split osteotomy. The fractures were located close to the former miniplates on the posterior or the anterior border in four cases. The new fracture occurred under the miniplate in only two cases. In these cases, there was only a single miniplate located on the external oblique line following Champy’s technique. This particular technique is recognized to protect the angle from tension force due to mastication [38], but not from force due to an anterior or a lateral impact. Following these observations, we hypothesized that miniplates reinforced the underlying bone, protecting it from fractures, and transmitted the forces to areas anterior or posterior to the miniplates. To support our hypothesis, we designed an experimental study consisting of simulating a mandibular trauma in order to compare the fractures caused by an impact on a mandible in the presence or absence of an internal fixation. We simulated an impact with a cylindrical 5 kg steel impactor at a speed of 7.4 m/s on the right parasymphysis region in 5 post mortem human subjects without miniplates on their mandible (control group) and in five post mortem human subjects bearing two miniplates on their right parasymphysis (miniplate group) [17]. In the control group, an impact simulated on the mandible without miniplates resulted in fractures with a simple, straight, and vertical line. These fracture lines could be related to the homogeneous distribution of the Von Mises stresses in our FEA model without miniplates. The same impact on the mandible bearing the same two miniplates as in our FEA model resulted in complex, tilted, multiple fracture lines, with comminution in several cases, at the anterior or posterior edge of the miniplate. These atypical fracture lines located at a distance from the surface impact at the edge of the miniplate could be explained by the Von Mises stress distribution shown in our FEA model with miniplates. These results supported our hypothesis on the transmission of forces.

Comminuted fractures occurring at the edges of the miniplates require more complex surgical procedures. Indeed, the presence of the former miniplates could affect the positioning of the new osteosynthesis miniplate; therefore, their removal may be required. However, removing an osseointegrated miniplate could be complex, and bone drilling may be necessary if the miniplate is covered by bone. Bone drilling might affect the new osteosynthesis miniplate’s positioning as well, and a longer miniplate might be required. Removing miniplates or using a longer miniplate may necessitate a longer incision and a larger subperiostal dissection. Furthermore, in the case of comminuted fracture or bone loss due to the former miniplates’ removal, load-bearing osteosynthesis miniplates should be used (rather than load-sharing miniplates). These considerations could account for the higher rate of postoperative complications such as malocclusion, postoperative infection, osteomyelitis, and malunion associated with recurrent mandibular fractures [7].

Our findings add further arguments in favour of the removal of miniplates in a specific sample of patients: those with a high risk of facial trauma. This population has been described in the literature [6,7,8,10,12,39]. It includes patients, mostly males, with a previous mandibular fracture occurring during a brawl or with a history of interpersonal violence, especially if associated with drug or alcohol addiction [6,10,11], patients practising contact or fighting sports [14,15], and soldiers (paratroopers and raiders) [16]. We recommend informing these patients of the risk of more complex fracture in the case of mandibular trauma with the perspective of delivering clear and fair information about the interest of miniplate removal.

Our study has further limitations. First, the temporomandibular joint’s anatomy was simplified, and the boundary conditions imposed on the condyles also resulted in a simplification of the temporomandibular joint’s movements. Consequently, the Von Mises stress distribution in the area of the condyles could not be described. These boundary conditions applied to the condyle can also influence the stress distribution, as certain loading scenarios, depending on their relative direction, can result in higher stress concentrations on the impact area.

Shared nodes between the screws and the bone simulated the osseointegration but may also facilitate the transmission of the forces through the screws to the bone compared with in vivo screwing.

The size of the mesh elements was constrained, in particular by the thinness of the miniplates. The maximal size possible was 1 mm. Thus, it was complicated to carry out a convergence analysis with a sufficient number of meshes with different element sizes without requiring the use of extremely small elements for a model of this size, leading to computational difficulties. Then, we decided to compare our model with a mesh including 1 mm elements to a smaller one with 0.5 mm elements. We did not find any major difference between the stress distribution and the maximal stress value.

Bones’ mechanical properties were simplified, retaining only a differentiation between the cortical bone and the cancellous bone and no correlation between these bones’ properties and Hounsfield units. Moreover, the thickness of the cortical bone was uniformly distributed (3 mm), which remained an approximation of the in vivo variable cortical thickness. Nevertheless, our two modelled mandibles shared the same bone mechanical properties. Our study focused on the fully consolidated bone after callus remodelling following fracture that resulted in “restitutio ad integrum” [40,41]. In this study, the isotropic material property assignment was adopted for the bone, like in most of the FEAs with a macroscopic scale [34,42,43,44,45]. However, the bone material is widely recognized as being anisotropic rather than isotropic [46]. This simplification of the bone material property may lead to a stiffer mandibular model [46].

Our finite element models simulate trauma on a fully dentate mandible with no third molars. Our results cannot be applied to other cases. In edentulous patients, the reduction in mechanical forces secondary to tooth loss results in changes in mandibular shape following bone resorption in less stimulated areas [47]. Moreover, tooth loss, by stimulating the osteoclastic activity, may be responsible for bone resorption along the alveolar crest [47]. Changes in the biomechanical properties of the mandibular bone or major changes in mandibular shape either due to a totally or partially edentulous dental arch may affect the stress distribution. The impact of mandibular miniplates in this population has not been studied yet. However, considering population ageing and the risk of fall in elderly patients [48], this situation should be assessed.

Age may also have an impact on the biomechanical behaviour of the mandible, bones’ mechanical properties, and mandibular shape. Indeed, the presence of an impacted third molar, depending on its position or angulation, alters the stress concentration in the mandibular angle [35], reducing the risk of condylar fractures but increasing the risk of angle fractures [36]. The presence of dental germs might favour mandibular fractures. Nevertheless, open reduction and internal fixation of these fractures are avoided in the presence of dental germs [49]. Biomechanical properties of bone are affected by the ageing process. The periosteum becomes thinner and cortical bone structure changes [50,51]. Mandibular shape evolves too, with the condyle becoming longer with mandibular growth [49]. In young patients, the biomechanical impact of the miniplates after bone healing has also not been studied yet. However, this issue should be addressed since these patients are likely to be exposed to a risk of recurrent fracture in the long term.

Furthermore, the soft tissues that could dampen or transmit impact forces were represented in none of our models. Therefore, the results of our comparison between the two models still remain valid. Moreover, despite these limitations which are usually encountered in FEAs, our results are in accordance with our clinical observations of mandibular fractures occurring on mandibles bearing miniplates [6] and with the results of our experimental study reproducing trauma on mandibles bearing miniplates [17].

Finally, there are only a few studies about the biomechanical impact of mandibular titanium miniplates after bone healing in the case of facial trauma recurrence in the literature, and those were mainly conducted by our research team. More clinical and experimental data are required to support our conclusion.

## 5. Conclusions

As suspected in further studies, our numerical simulation tends to confirm that the presence of miniplates on the mandible may cause more complex fractures in the case of trauma. Consequently, we recommend highlighting this risk in the information delivered to patients about the interest of removing miniplates after bone consolidation in patients with previous assault injuries, patients suffering from addictions (alcohol or substance abuse), patients practicing fighting or contact sports, or those involved in military activities.

In the future, our model will be improved to study the strains of the whole mandible, including the condyles. In this way, we need to free the mandible from the boundary conditions applied to the condyles. To achieve this, we will incorporate in the model the temporomandibular joint soft tissues (capsule, disc, and synovial liquid) and possibly the soft tissues of the face. Moreover, the biomechanical properties of bone will be correlated to the Hounsfield units to be more precise. Eventually, to validate our proposal for the removal of titanium osteosynthesis in patients with a high risk of mandibular fracture, a prospective, randomized clinical study comparing a group of patients at risk of trauma retaining miniplates and a group of patients at risk not retaining miniplates should be conducted. This would enable us to assess the incidence of mandibular fracture recurrences and the complications of a systematic osteosynthesis removal versus the complexity of the osteosynthesis of a new fracture at the edge of a miniplate, along with the socio-economic impact and the impact on the healthcare system.

## Figures and Tables

**Figure 1 bioengineering-10-00994-f001:**
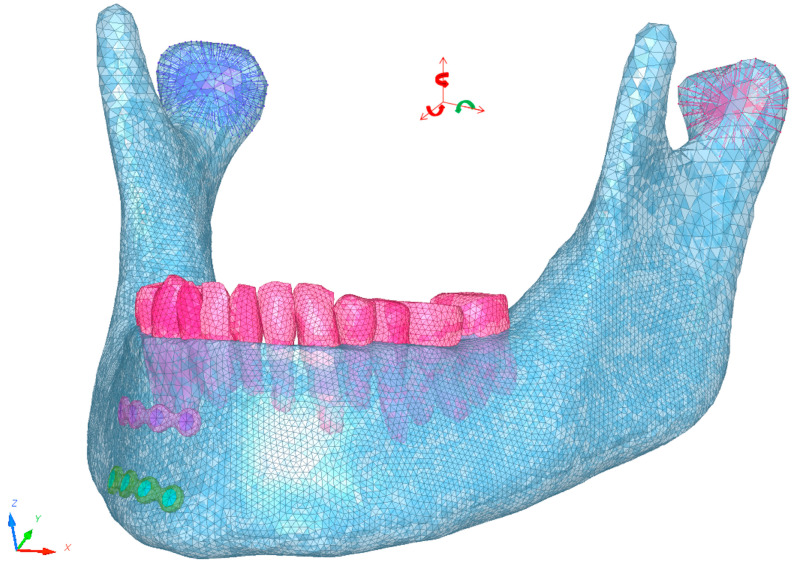
Boundary conditions applied to the condyles. Translation blocked in all three directions (red straight arrows). Rotation blocked in axial and frontal plane (red curved arrows) and authorized in sagittal plane to reproduce mouth opening and closing (green curved arrows).

**Figure 2 bioengineering-10-00994-f002:**
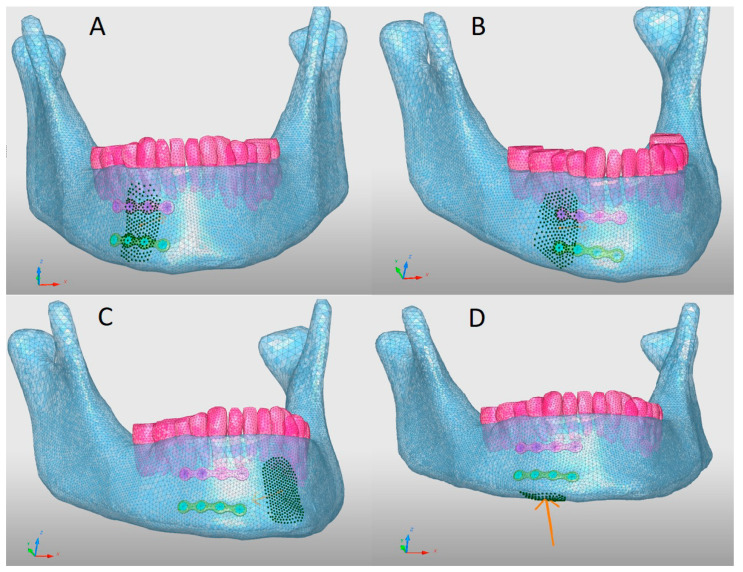
Concentrated loads of 2000 N applied to a mandible bearing parasymphyseal miniplates: (**A**) on the parasymphysis; (**B**) on the area straddling the edge of the plates and the adjacent bone; (**C**) at a distance from the miniplates on the symphysis; (**D**) on the basilar border below the miniplates (the orange arrow shows the direction of impact).

**Figure 3 bioengineering-10-00994-f003:**
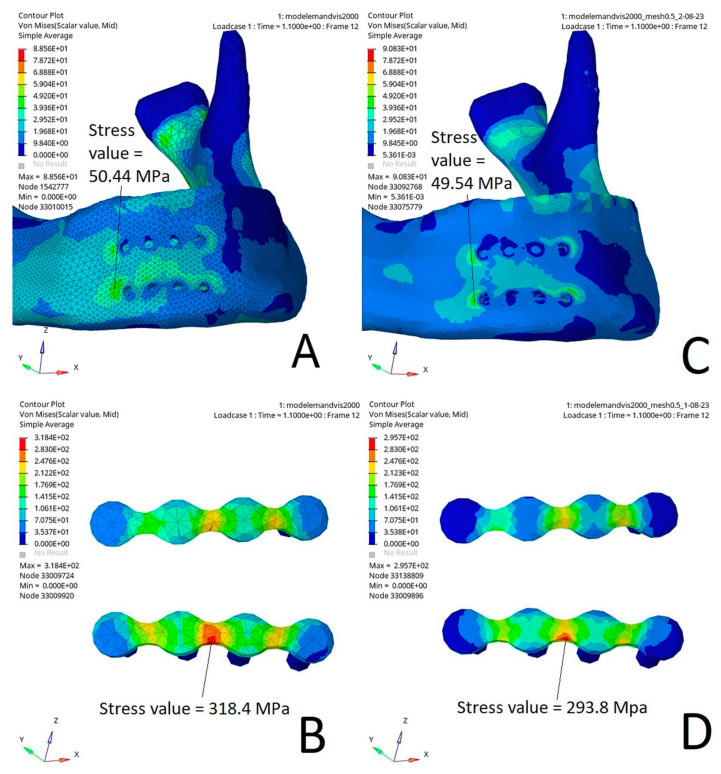
Comparison of the Von Mises stress distribution (MPa) in a direct impact on the parasymphysis using a tetrahedral mesh with 1 mm elements ((**A**) mandible, (**B**) osteosynthesis) and a tetrahedral mesh with 0.5 mm elements ((**C**) mandible, (**D**) osteosynthesis). The size of the tetrahedrons did not change the Von Mises stress distribution in the area of interest near the miniplates.

**Figure 4 bioengineering-10-00994-f004:**
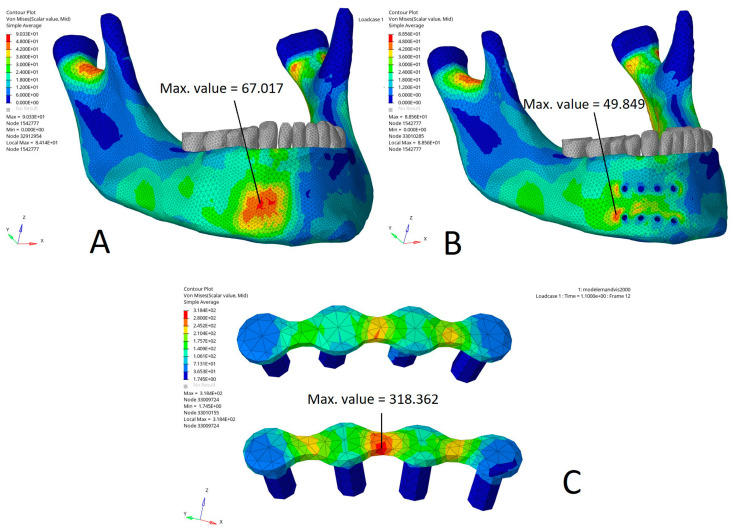
Von Mises stress distribution (MPa) in a direct impact on the parasymphysis: (**A**) model without miniplates, maximal stress area superimposed upon the impact area; (**B**) model with miniplates, two maximal stress areas on the posterior border of the mandible bearing two miniplates (on the cortical bone at a distance from the impact); (**C**) impact area, stresses mainly applied to the miniplates. Max. value: maximal stress value (MPa) in the area of interest, i.e., in the vicinity of the impact and the miniplates (the maximal stress values in the condylar region have deliberately not been taken into account because they are close to the boundary conditions and thus are difficult to interpret, as they could be influenced by the boundary conditions).

**Figure 5 bioengineering-10-00994-f005:**
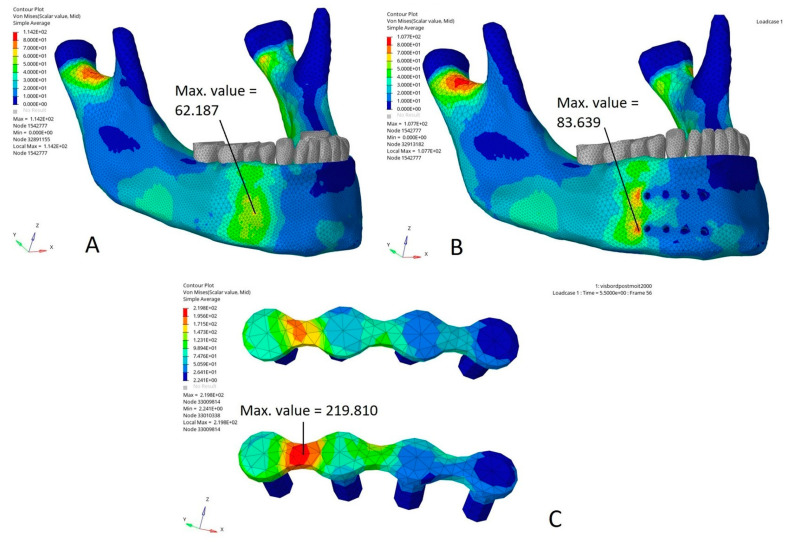
Von Mises stress distribution (MPa) in a direct impact on the mandibular corpus: (**A**) model without miniplates, maximal stress area superimposed upon the impact area; (**B**) model with miniplates, two maximal stress areas (on the cortical bone at the posterior border of the two miniplates); (**C**) impact area, stresses mainly applied to the miniplates. Max. value: maximal stress value (MPa) in the area of interest, i.e., in the vicinity of the impact and the miniplates (the maximal stress values in the condylar region have deliberately not been taken into account because they are close to the boundary conditions and thus are difficult to interpret, as they could be influenced by the boundary conditions).

**Figure 6 bioengineering-10-00994-f006:**
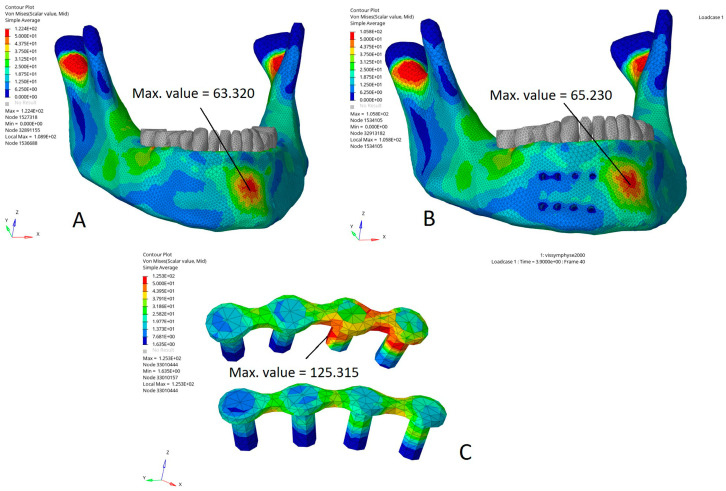
Von Mises stress distribution (MPa) in a direct impact on the symphysis: stress distributions relatively similar in the model without miniplates (**A**) and in the model with parasymphyseal miniplates (**B**). Stresses mainly applied to the miniplates and the fixation screw area in the cortical bone (**C**). Max. value: maximal stress value (MPa) in the area of interest, i.e., in the vicinity of the impact and the miniplates (the maximal stress values in the condylar region have deliberately not been taken into account because they are close to the boundary conditions and thus are difficult to interpret, as they could be influenced by the boundary conditions.).

**Figure 7 bioengineering-10-00994-f007:**
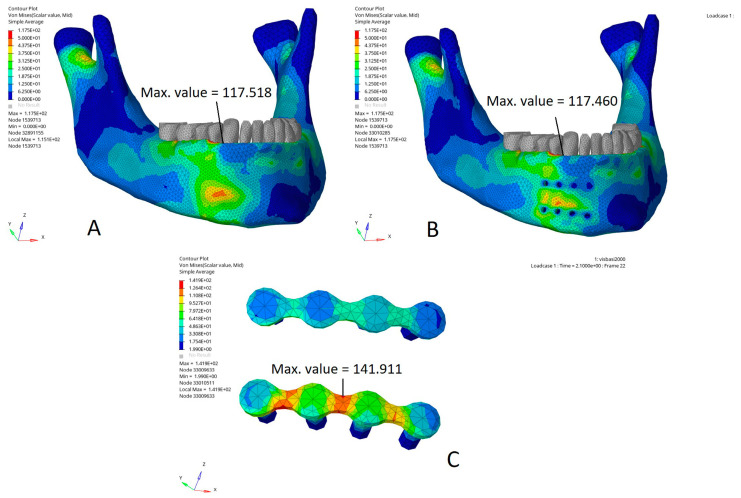
Von Mises stress distribution (MPa) in an impact on the basilar border: (**A**) model without miniplates, single maximal stress area halfway up the mandible; (**B**) model with parasymphyseal miniplates, several maximal stress areas (on the cortical bone at the posterior border of the basilar miniplate, and between the two miniplates); (**C**) impact area, stresses mainly applied to the basilar miniplate and the fixation cortical screws. Max. value: maximal stress value (MPa) in the area of interest.

**Figure 8 bioengineering-10-00994-f008:**
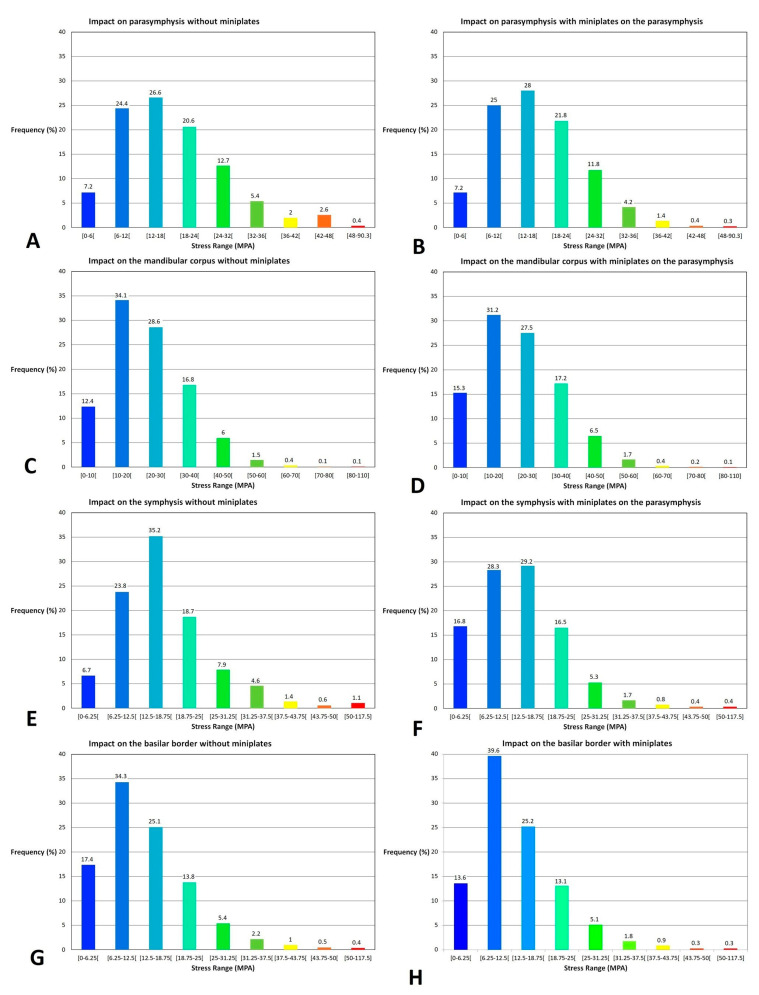
Range stress histograms: (**A**) direct impact on the parasymphysis in a model without miniplates; (**B**) direct impact on the parasymphysis in a model with miniplates on the parasymphysis; (**C**) direct impact on the mandibular corpus in a model without miniplates; (**D**) direct impact on the parasymphysis in a model with miniplates on the parasymphysis; (**E**) direct impact on the symphysis in a model without miniplates; (**F**) direct impact on the symphysis in a model with miniplates on the parasymphysis; (**G**) impact on the basilar border in a model without miniplates; (**H**) impact on the basilar border in a model with miniplates on the parasymphysis.

**Figure 9 bioengineering-10-00994-f009:**
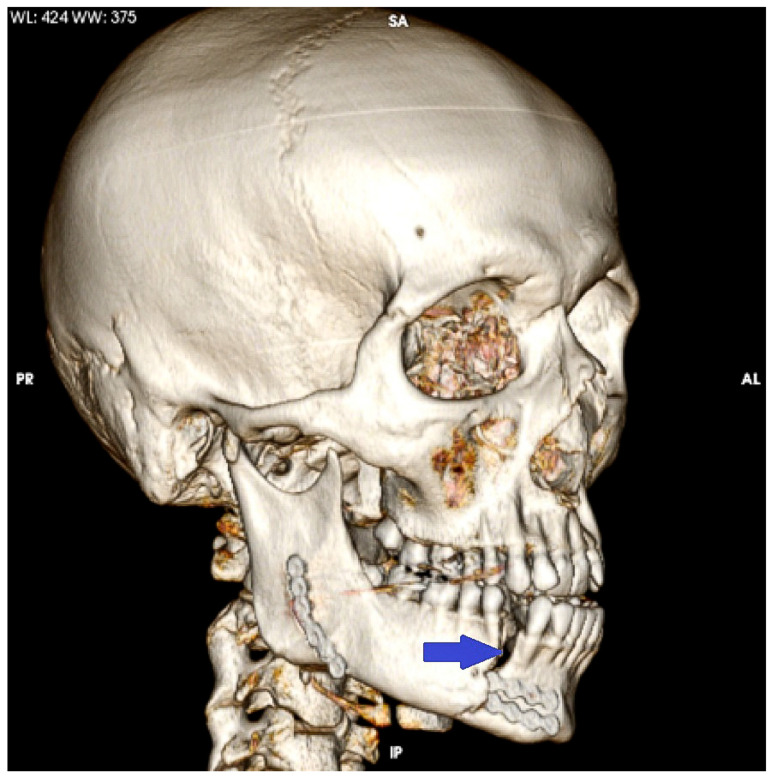
Three-dimensional reconstruction of a CT scan, fracture of the mandibular corpus (blue arrow) at the posterior border of two titanium miniplates.

**Table 1 bioengineering-10-00994-t001:** Material property settings applied in our finite element model.

	Young’s Modulus (MPa)	Poisson ‘s Ratio	Initial Density (g/cm^3^)
Compact bone	13,000	0.3	1.85
Cancellous bone	56	0.3	1.5
Teeth	18,600	0.31	1.8
Titanium	114,000	0.34	4.5

## Data Availability

The original contributions presented in the study are included in the article, further inquiries can be directed to the corresponding author.

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
