# Peer review of "Mandibular Titanium Miniplates Change the Biomechanical Behaviour of the Mandible in the Case of Facial Trauma: A Three-Dimensional Finite Element Analysis"

_bioengineering, 2023, doi:10.3390/bioengineering10090994_

Round 1
Reviewer 1 Report
The authors provided in the paper the results of a study looking at the biomechanical behavior of mandibles with or without titanium miniplates through the Finite Element Analysis.
They accurately described the model that has been used and then they simulated 4 different trauma areas.
I really appreciated the clearly images they provided which immediately clarify the concept expressed.
Eventually, in the discussion section they correctly highlighted the limitations of the study.
Suggestions:
- In the end of the discussion section previous clinical studies in accordance with the data of the current study are just mentioned. I suggest going into detail a little more.
- Given the limitations of the current study that have been mentioned, I suggest adding in the conclusions the future perspective to improve the quality of the evidence.
Author Response
Dear Reviewer,
On behalf of my co-authors, I would like to thank you for your valuable comments about our manuscript.
We improved our manuscript, taking into account all your comments:
- In the end of the discussion section previous clinical studies in accordance with the data of the current study are just mentioned. I suggest going into detail a little more.
We described more in detail in the discussion our previous clinical and experimental studies:
In this multicenter case series, we identified 13 patients presenting a fracture on a mandi-ble bearing miniplates for at least 6 months. The indication of the former osteosynthesis was a fracture due to assault in nine cases, a horse hoof kick in one case, and a bilateral sagittal split osteotomy for correction of dental class II in three cases. The mechanism of the second trauma was assault in 10 cases, including the 9 cases who initially presented with a fracture due to an assault, a subsequent hoof kick for the patient who had been kicked initially, and a traffic accident for 2 patients treated with bilateral sagittal split os-teotomy. The fractures were located close to the former miniplates on the posterior or the anterior border in four cases. The new fracture occurred under the miniplate in only two cases. In these cases, there was only a single miniplate located on the external oblique line following the Champy’s technique. This technique protects the angle from tension force due to mastication [37], but not from force due to an anterior or a lateral impact. Then, we hypothesized that miniplates reinforced the underlying bone, protecting it from fractures, and transmitted the forces to areas anterior or posterior to the miniplates. To support our hypothesis, we designed an experimental study, consisting in simulating a mandibular trauma in order to compare the fractures caused by an impact on the mandible in the presence or absence of an internal fixation. We simulated an impact with a cylindrical 5 Kg steel impactor at a speed of 7.4 m/s on the right parasymphysis region in 5 post-mortem human subjects without miniplate on their mandible (control group) and in five post-mortem human subjects bearing two miniplates on their right parasymphysis (miniplate group) [17]. In the control group, an impact simulated on a mandible without miniplates, resulted in fractures with a simple, straight, and vertical line. These fracture lines could be related to the homogeneous distribution of the Von Mises stresses in our FEA model without miniplates. The same impact on the mandible bearing the same two miniplates as in our FEA model, resulted in complex, tilted, multiple fracture lines, with comminution in several cases, at the anterior or posterior edge of the miniplate. These atypical fracture lines located at a distance from the surface impact at the edge of the miniplate could be explained by the Von Mises stress distribution shown in our FEA model with miniplates. This results supported our hypothesis on the forces transmission.
- Given the limitations of the current study that have been mentioned, I suggest adding in the conclusions the future perspective to improve the quality of the evidence.
We described our future perspective in the conclusion:
“In the future, our model will be improved to study the strains of the whole mandible, in-cluding the condyles. In this way, we need to free the mandible from the boundary condi-tions applied to the condyles. To achieve this, we will incorporate in the model the tem-poro-mandibular joint soft tissues (capsule, disc and synovial liquuid), and possibly the soft tissues of the face. Moreover, the biomechanical properties of bone will be correlated to the Hounsfield units to be more precise. Eventually, to validate our proposal for removal of titanium osteosynthesis in patients with a high risk of mandibular fracture, a prospec-tive, randomized clinical study comparing a group of patients at risk of trauma retaining miniplates and a group of patients at risk not retaining miniplates should be conducted. It would enable us to assess the incidence of mandibular fracture recurrences, the complica-tions of a systematic osteosynthesis removal versus the complexity of the osteosynthesis of a new fracture at the edge of a miniplate, along with the socio-economic impact, and the impact on the healthcare system.”
I hope we were able to meet your expectations.
Yours faithfully.
Reviewer 2 Report
|
Need to be improvised
Author Response
Dear Reviewer,
On behalf of my co-authors, I would like to thank you for your valuable comments about our manuscript.
Please find below our answers to your comments:
Title
I suggest- Biomechanical behaviour of the mandible fracture treated with titanium miniplates: A three-dimensional finite element analysis.
The paper is not focused on the treatment (or the biomechanical behavior) of mandibular fracture with titanium miniplates. The topic is the biomechanical change due to the conservation of the miniplates on a mandible AFTER bone healing compared to a mandible with miniplate removal after bone healing in case of trauma.
In this study, we discuss the interest of miniplate removal after bone healing.
- 2. Abstract
- Introduction
- a) It is not clear what type of mandible fractures the authors are focusing on. It is better to focus on one type of fracture, ie. Location (Vertical/horizontal/combined fracture) or morphology (simple, linear, complex etc). I suggest authors should use AO classifications to determine and report this fracture in the introduction part.
The paper does not deal with mandibular fracture. In our study, we consider that the patient had a healed fracture of the right parasymphysis of the mandible, corresponding to a fully consolidated bone after callus remodelling following fracture resulted in “restitutio ad integrum” (as described in the literature regarding bone biology: Bahney et al.. Cellular Biology of Fracture Healing. J. Orthop. Res. Off. Publ. Orthop. Res. Soc. 2019, 37, 35–50, Marsell et al.The Biology of Fracture Healing. Injury 2011, 42, 551–555.). The use of miniplates and their position indicate that the previous fracture, currently healed, was a simple vertical fracture of the right parasymphysis.
This was added to the manuscript:” Two 4-hole 1,0 mm thick 2.0 miniplates, (Global D, Brignais, France) were placed on the right parasymphysis simulating the previous treatment of a vertical simple parasym-physis fracture. We considered that the patient had a healed fracture of the right para-symphysis of the mandible, corresponding to a fully consolidated bone after callus re-modelling following fracture resulted in “restitutio ad integrum”.
Then, we considered that this fracture was successfully treated with an internal fixation with two miniplates. In clinical practice, in some cases, the miniplate are removed after bone healing, and in some other cases, the miniplates are left in place. That is the subject of our comparison. So at the time of our analysis, the bone is fully restaurate (“restitutio ad integrum”), and the only difference between the two models is the presence or the absence of miniplates.
We discuss in this study the biomechanical consequences of the conservation of miniplates in case of facial trauma.
- b) Authors should report the failure of treatment of mini plates in clinical practice. This will determine the reason why you chose the fracture
There is no failure of the treatment. The treatment was a success and the bone healed well with the internal fixation. However, facial trauma recurrence is frequent. Then, if a patient presents a facial trauma recurrence on a mandible with miniplates still present on a healed bone, one question deserves to be asked: what are the biomechanical consequences? No study tried to give an answer to this question in the literature, except our experimental study (Graillon et al.. Do Mandibular Titanium Miniplates Affect the Biomechanical Behaviour of the Mandible? A Preliminary Experimental Study. J. Stomatol. Oral Maxillofac. Surg. 2022, S2468-7855(22)00038-6).
Mandibular fractures after an impact on a healed mandible fracture with miniplates still present exist in clinical practice and has been described in the literature (Graillon et al. Do Mandibular Miniplates Increase the Risk of Complex Fracture in Facial Trauma Recurrence? Case Series. J. Cranio-Maxillofac. Surg. 2020; Elrasheed et al. Mandibular Fractures That Have Healed Are Not Weakened Permanently: Series of Nine Patients Who Sustained Mandibular Fractures at Different Sites on Two Separate Occasions. Br. J. Oral Maxillofac. Surg. 2011, 49, 209–212; Agir et al. Fracture Patterns and Bone Healing in Recurrent Mandibular Fractures: A Clinical Study of 13 Patients. Plast. Reconstr. Surg. 2005, 116, 427–436; discussion 437-439)
- c) Authors should report the drawbacks of the previously reported literature. This will lead to research gap findings. Thus, it is easier to plan our research study and framework.
Miniplate removal after bone healing is controversial. The arguments to not perform a removal of the miniplates after bone healing, is based on economics and biological considerations Brown et al.The Fate of Miniplates in Facial Trauma and Orthognathic Surgery: A Retrospective Study. Br. J. Oral Maxillofac. Surg. 1989, 27, 306–315. Bhatt et al. Removal of Miniplates in Maxillofacial Surgery: A Follow-up Study. J. Oral Maxillofac. Surg. Off. J. Am. Assoc. Oral Maxillofac. Surg. 2005, 63, 756–760. Mosbah et al Miniplate Removal in Trauma and Orthognathic Surgery--a Retrospective Study. Int. J. Oral Maxillofac. Surg. 2003.) Biomechanical consequences has never been evaluated nor taken into account in this debate.
- d) Para for FEA. It is very limited information about it. No important point from here. It should be highlighting the previous FEA study on mandibles treated with mini plates so that we can see the research gap clearly.
Previous FEA, only focused on stability of osteosynthesis before bone consolidation. Nobody described the biomechanical behavior of the miniplate after bone healing, which could be an argument to propose a systematic removal.
- e) The most concern for me is the research gap. Somehow related to the problem statement. What is the importance of doing this study? This should have come from the gap of research. Please highlight more about this.
The study will give a biomechanical argument in the debate about the systematic removal of the miniplates. No study has ever focused on this biomechanical point of view.
Do we chose to not remove miniplates because it’s less expansive for the health system, even if it exposes our patients to a risk of complex mandibular fracture which can impaired their mastication for the rest of their life?
- Materials and Methods
- a) Even though the authors used the previous FE model, it is suggested that the mesh size should be reported in this section too.
Mesh size was 1 mm.
- b) I am not sure whether the authors have done with convergence study or not. Since in this manuscript, there is no convergence study reported. If no convergence study, so what is the element size used by authors in their FEA? This is crucial to report.
We did not carry out a mesh sensitivity or a convergence study. We used a fine mesh of 1 mm.
- c) Cortical and cancellous were not reported in this section. I am aware that the authors have reported this in their previous publications, but for me, it should be again mentioned in this manuscript and do citation.
We do not understand what you mean by “cortical and cancellous bone were not reported in this section”. Material properties of cortical and cancellous bone are reported in the materials and methods section in the table 1.
- d) What is the means of 3mm-thick cortical tetrahedral 3D meshing? Is it tetrahedral with 3mm mesh size, or the 3mm is for the cortical’s thickness? If cortical’s thickness, is it uniformly distributed through all cortical region? And how you segment the cortical up to 3mm thick? Please justify.
We create a cortical with a thickness of 3mm. The meshing of the cortical bone used tetrahedral elements.
The 3 mm thickness of the cortical was uniformly distributed in the whole mandible, which is an approximation, similar in the two models (without and with miniplates) compared. This limitation has been added in the discussion section. We choose a thickness of 3 mm, because it corresponded to the thickness of the mandible in the area of the parasymphysis measured on the initial CTscan.
- e) Subtopic Boundary Conditions should be in Italic font.
Boundary condition has been written in italic font.
- f) Authors should put some figures that showing the boundary (fixed and applied load). So that readers would understand it more clearly. The statement from the sentence is not sufficient to make readers understand. If the readers understand, high chances to cite this work/manuscript in the future.
The figure has been added (figure 1).
- g) Material properties. The model was assigned with homogeneous properties. Normally, homogeneous properties as used by the authors leads to an overestimation of organ stiffness. See J Biomech, 46 (2013), 2710-2721; and Com Meth Appl Mech Eng, 198 (2009), 2167-2217. Please clarify on this method.
Indeed, the materials are heterogeneous, but due to a lack of data in the literature, we decided to simplify with homogeneous data, as frequently proposed in the literature (Kılınç, et al. Computer Methods in Biomechanics and Biomedical Engineering 21 (7): 488‑97. Liu et al.,. Journal of Zhejiang University. Science. B 19 (1): 38‑48. Santos et al. Journal of Oral and Maxillofacial Surgery: Official Journal of the American Association of Oral and Maxillofacial Surgeons 73 (4): 683‑91. Antic et al. International Journal of Oral and Maxillofacial Surgery 45 (5): 588‑92. Gallas Torreira et al. Journal of Cranio-Maxillo-Facial Surgery: Official Publication of the European Association for Cranio-Maxillo-Facial Surgery 32 (5): 303‑7.).
In our perspectives, we propose to adapt Young's modulus to bone densities on CT scan (HU)..
- h) Authors used isotropic properties. It should be discussed why the assumption of isotropy makes sense in beam-type structures and also which components of the actually anisotropic stiffness tensor enter the fictitiously isotropic material description. See example: J Biomech, 46 (2013), 2710-2721; Med Eng Phys, 30 (2008), 924-930; and Med Eng Phys, 28 (2006), 227-233.
Bone is an isotropic material. Also due to a lack of data on the macroscopic scale in the literature, we made an approximation by considering it to be isotropic, as in most of other mandibular FEAs in the literature (Wieja et al. 2021, Huang et al. 2021, Kargarnejad et al. 2021, Zhong et al. 2021 Savoldelli et al. 2021, WĄdoŁowski, KrzesiŃski, et Gutowski 2020, da Silveira et al. 2021, Pavlychuk et al. 2020, Chang et al. 2020, Pałka et al. 2020, Guastaldi et al. 2020, Ramos, Semedo, et Mesnard 2020, Arora et al. 2020, B. Park et al. 2020).
This limitation has been added and highlighted in the discussion.
Nevertheless, you quote 3 papers, 2 of them described a FEA of a small part of the femur of a mouse, and the last one on human femur concluded: “the isotropic simplification does not show much difference from the orthotropic material property assignment under the two specific loading conditions”. The simplification of the material property of the bone remains acceptable to us
- Tetrahedral elements were used. However, higher accuracy of analysis can be achieved by utilizing the hexahedral element to the bone. As compared with tetrahedral element, hexahedral was in theoretical have better accuracy due to its nature of having relatively more degrees of freedom. See Com Meth Appl Mech Eng, 189 (2000), 961-974. Please justify your choices of setting tetrahedral in the model.
The geometric complexity of the model does not allow automatic meshing in hexahedrons with a reasonable element size to be carried out quickly and cleanly.
The tetrahedron is frequently used in human body modelling (for example in your previous reference Med Eng Phys, 30 (2008)), although it is less accurate than the hexahedron. In the future, the mesh could be refined and transformed into a hexahedron.
- j) There are many bodies in the FE model, but how authors define the contact condition between the teeth and bone, plate and bone, cancellous and cortical, cortical and plate, cancellous and plate, teeth and cortical, and teeth and cancellous. Authors should put the friction coefficient too.
Anatomical parts in contact were meshed in a continuous mesh. For other contacts, we used a penalty contact to avoid penetration of one object by another. Then, no friction coefficient were necessary.
This was added to the manuscript:
“Contact conditions
The mesh was continue between the teeth and bone, cancellous and cortical bone, cortical bone and miniplate, cancellous bone and screws. Penalty contacts were applied between mandibular and maxillar teeth and between mandible and facial bone to avoid penetration of one part by another.”
- k) Validation issue. The authors said the the FE model was taken from their previous study, but not sure how is the validation study. Even though, if the authors have stated in the previous literature, the authors should again report the validation of the FE model in this manuscript. If not, verification is sufficient. It should be noted that the FE model should be validated prior to any parametric study. See example Materials, 16(9), 2023, 3298 and Injury 53 (7), 2022, 2424-2436.
The model has been previously validated in the literature with tests by Schneider, and Viano.
The model was validated according to the following:
- Experimental trials conducted by Schneider et al. (Impact studies of facial bones and skull Proceedings 16th Stapp Car Crash Conference, SAE Paper, vol. 720965 (1972), pp. 186-203.) The resistance of the facial bones and mandible was assessed via 106 trials on 17 PMHS skulls. Two different positions were used to impact the mandibles: antero-posterior and lateral. Two Plexiglas cross members were affixed to the drop assembly and contained nylon bushings to minimise any frictional reaction with the steel guidewires. In addition, the heads were severed at the seventh cervical vertebra in the latter experiments to facilitate the placement of the skull necessary for antero–posterior mandibular impacts. The head, in all cases, was supported by wedges of soft polyurethane padding
- Experimental trials conducted by Viano et al. (Concussion in professional football: comparison with boxing head impacts – part 10 Neurosurgery, 57 (December (6)) (2005), pp. 1154-1172) where the effect of a boxer's punch was studied: eleven Olympic boxers weighing 51 kg (112 lb)–130 kg (285 lb) were included in the study. These boxers were instructed to strike the instrumented Hybrid III head with their gloved fist two times with four different punches: a straight punch to the forehead, a straight punch to the jaw, a hook, and an uppercut. Accelerometers were placed in the boxer's clenched hand (two Endevco 7264-2k accelerometers). The Hybrid III was equipped with the standard triaxial accelerometers (Endevco 7264-2k).
- Experimental trials described previously in this study.
This was added to the manuscript in the material and methods section.
“The model was previously validated [20] according to experimental trials conducted by Schneider et al. [21], Viano et al. [22], and by experimental study in the Tuchtan et al. [20] study itself.”
- l) Please provide ethical approval number. This is to ensure that the DICOM data is protected under the ethics committee. If not, then the study is illegible.
This is a retrospective and anonymous data in a study not involving human being, in accordance with the law: LOI n° 2012-300 du 5 mars 2012 relative aux recherches impliquant la personne humaine https://www.legifrance.gouv.fr/loda/id/JORFTEXT000025441587/, this study did not require an ethical approval. For example, the paper you quote previously, Materials, 16(9), 2023, 3298 also did not provide any ethical approval, Institutional Review Board Statement, or Informed Consent Statement.
- Results
- a) Von mises stress. Why authors select this? Since the one is not steel, but consists of collagen-reinforced mineral polycrystals, the Mohr-Coulomb criterion, including version for upscaling from the mineral to the issue observation scale, was shown to be probably much more relevant for the fracture characteristics of bone. See example J Theor Biol, 260 (2009), 230-252; and Int J Plast, 91 (2017), 238-267. I think the authors are missing so much information and discussion about this. Some limitations should be explained and justified with more theoretical and relate with previous published literature so that the results will not overestimate.
Our choice to use Von Mises is partly linked to what is done in the literature (for example the papers you quote previously Materials, 16(9), 2023, 3298 and Injury 53 (7), 2022, 2424-2436) or in biomechanics, where Von Mises stresses are very often used. Moreover, in the context of our study, we are interested in stress distribution rather than fracture aspects.
- b) Figure 1. I cannot see where the fracture side is. If no fracture, then this study is not valid. For me, you put the mini-plates meaningless if no fracture was simulated in this study.
The aim of this study was not to simulate a bone fracture but to simulate an impact on a healed mandible with miniplates, to evaluate the biomechanical behavior of a mandible bearing miniplate after bone healing.
- c) Figure 2, 3, 4 and 5. The scale of stress (left part) is overlap with the contour plot picture of the model. Please correct it.
It has been corrected.
- d) Figure 2, 3, 4 and 5. The scale is small, cannot see it clearly.
It has been improved.
- e) Figure 2, 3, 4 and 5. I suggest putting and label maximum stress on the figure. See example Materials, 16(9), 2023, 3298 and Injury 53 (7), 2022, 2424-2436.
It has been added.
- f) I think presenting stress results only is not sufficient for this high-quality journal (Bioengineering is a Q2 ISI ranking in its field). I would suggest authors to add more results (ie. Displacement, strain, pin-bone stress, contact pressure, micromovement at the fracture side, and many more) so that the manuscript will be more insightful to the readers and create high chances to be cited by other researchers in the future.
We looked exclusively at stress distribution, because our aim was to evaluate how stresses are distributed during impact on a healed mandible with miniplates, and whether there are stress concentrations around the miniplates, which could result in a weakening in this area during impact. As there is no fracture, we do not expect any displacement, except for the mouth opening, allowed by the boundary conditions, depending of the impact.
- g) To make more interested results, I would suggest authors to include stress histogram. So we can see where is the location for high frequency of stress.
We do not exactly understand what you are waiting for. Our figures seemed to us particularly clear and simple.
- Discussion
- a) Very limited discussion about the results and comparison between author’s results with previous published literature by others. Authors should compare, at least find the difference between the stress from others and their own.
This is the first study dealing with this topic. The first description of the biomechanical impact of mandibular miniplates after bone healing. The first biomechanical argument in the debate regarding the miniplates removal after mandibular bone healing. Previous published literature mostly described the biomechanical behavior of internal fixation on a non-healed mandible fracture. Their goal is to evaluate the stability of the osteosynthesis. Comparing the biomechanical behavior of an osteosynthesis with an underlying fracture versus an osteosynthesis with an underlying healed bone would be meaningless. In one case we want to be sure that the osteosynthesis is sufficiently stable to allow bone healing, in the other case we want to know if the presence of the osteosynthesis changes the biomechanical behavior of the mandible and expose to a risk of more complex fracture in case of trauma recurrence.
- b) More limitation of study should be included here. I can see only one, but this is not enough. The limitation is actually including; boundary conditions, material properties, contact conditions between different bodies, selection of stress results, and many more.
Please find attached the limitations we described, highlighted parts has been added:
“Our study has further limitations. First, the temporomandibular joint’s anatomy was simplified and the boundary-conditions imposed to the condyles also resulted in a sim-plification of the temporomandibular joint’s movements. Consequently, the Von Mises stress distribution in the area of the condyles could not be described. This boundary-conditions applied to the condyle can also influence the stress distribution, as certain loading scenarios depending on their relative direction can result in higher stress concentrations on the impact area.
Shared nodes between the screws and the bone simulated the osseointegration, but may also facilitate the transmission of the forces through the screws to the bone compared with in vivo screwing.
Bones’ mechanical properties were simplified retaining only a differentiation between the cortical bone and the cancellous bone, and no correlation between these bones’ properties and Hounsfield units. Moreover, the thickness of the cortical was uniformly distributed (3mm), which remained an approximation of the in vivo variable cortical thickness. Nevertheless, our two modelled mandibles shared the same bone mechanical properties. Our study focused on the fully consolidated bone after callus remodelling following fracture resulted in “restitutio ad integrum” [39,40]. In this study, the isotropic material property assignment was adopted for the bone, like in most of the FEA with a macroscopic scale [34,42–45]. However, the bone material is widely recognized as being anisotropic rather than isotropic [46]. This simplification of the bone material property may lead to a stiffer mandibular model [46].
Our finite element models simulate a trauma on a fully dentate mandible with no third molars. Our results cannot be applied to other cases. In edentulous patients, the reduction of mechanical forces secondary to tooth loss, results in changes in mandibular shape following bone resorption in less stimulated areas [41]. Moreover, tooth loss, by stimulating the osteoclastic activity may be responsible for bone resorption along the alveolar crest [41]. Changes in the biomechanical properties of the mandibular bone or major changes in mandibular shape either due to a totally or partially edentulous dental arch may affect the stress distribution. The impact of mandibular miniplates in this population has not been studied yet. However, considering population ageing and the risk of fall in elderly patients [42], this situation should be assessed.
Age may also have an impact on the biomechanical behaviour of the mandible, bones’ mechanical properties and mandibular shape. Indeed, the presence of an impacted third molar, depending on its position or angulation, alters the stress concentration in the mandibular angle [33], reducing the risk of condylar fractures but increasing the risk of angle fractures [34]. The presence of dental germs might favour mandibular fractures. Nevertheless, open reduction and internal fixation of these fractures are avoided in the presence of dental germs [43]. Biomechanical properties of bone are affected by the aging process. The periosteum becomes thinner and cortical bone structure changes [44,45]. Mandibular shape evolves too, with the condyle becoming longer with mandibular growth [43]. In young patients, the biomechanical impact of the miniplates after bone healing has also not been studied yet. However, this issue should be addressed since these patients are likely to be exposed to a risk of recurrent fracture in the long term.
Furthermore, the soft tissues which could dampen or transmit impact forces were rep-resented in none of our models. Therefore, the results of our comparison between the two models still remain valid. Moreover, despite these limitations which are usually en-countered in FEA, our results are in accordance with our clinical observations of man-dibular fractures occurring on mandibles bearing miniplates [6], and with the results of our experimental study reproducing trauma on mandibles bearing miniplates [17].
Finally, there are only a few studies about the biomechanical impact of mandibular titanium miniplates after bone healing in case of facial trauma recurrence in literature, and mainly conducted by our research team. More clinical and experimental data are required to support our conclusion.”
I hope we were able to meet your expectations.
Yours faithfully.
Reviewer 3 Report
The authors perform a finite element analysis aiming to investigate the influence of titanium miniplates on the biomechanical behavior of human mandibles subjected to impacts.
The manuscript is well-written and organized.
Materials and methods are clearly described.
The study design seems to be adequate to achieve the purpose of the paper since different types of trauma simulations were performed and the mandibular model was well designed.
The conclusions are well-supported by the results.
Minor comments follow:
Line 86: Italic characters should be used.
Figures
Von Mises colored scales with values are hardly visible because superimposed on the mandible images. Please, modify to increase readability.
Figure legends are textual. Please, modify them in a more didascalic mode. e.g, "Figure 2. illustrates the Von Mises stress distribution (MPa) in a direct impact on the parasymphysis: A. In the model without miniplates showing a maximal stress area superimposed upon the impact area, B. In the model with miniplates, showing two maximal stress areas located on the cortical bone at a distance from the impact, on the posterior border of the mandible bearing two miniplates, C. On the impact area, the stresses were mainly applied to the miniplates."
change to:
"Figure 2. Von Mises stress distribution (MPa) in a direct impact on the parasymphysis: A. Model without miniplates, maximal stress area superimposed upon the impact area. B. Model with miniplates, two maximal stress areas on the posterior border of the mandible bearing two miniplates (on the cortical bone at a distance from the impact). C. Impact area, stresses mainly applied to the miniplates."
Apply the same principle also for the other figure legends.
Discussion
Line 235: "loud -sharing" should be "load-sharing"
There are only a few studies about this topic in literature (and those cited are by the authors themselves). This limitation, together with the need for more clinical data to support the authors' conclusions, should be mentioned.
The English language seems fluent and properly structured.
Author Response
Dear reviewer,
On behalf of my co-authors, I would like to thank you for your valuable comments.
We improve our manuscript, taking into account all yours comments:
- Line 86: Italic characters should be used.
Italic characters has been used.
Figures
- Von Mises colored scales with values are hardly visible because superimposed on the mandible images. Please, modify to increase readability.
Figures has been modified as requested.
- Figure legends are textual. Please, modify them in a more didascalic mode. e.g, "Figure 2. illustrates the Von Mises stress distribution (MPa) in a direct impact on the parasymphysis: A. In the model without miniplates showing a maximal stress area superimposed upon the impact area, B. In the model with miniplates, showing two maximal stress areas located on the cortical bone at a distance from the impact, on the posterior border of the mandible bearing two miniplates, C. On the impact area, the stresses were mainly applied to the miniplates."
change to:
"Figure 2. Von Mises stress distribution (MPa) in a direct impact on the parasymphysis: A. Model without miniplates, maximal stress area superimposed upon the impact area. B. Model with miniplates, two maximal stress areas on the posterior border of the mandible bearing two miniplates (on the cortical bone at a distance from the impact). C. Impact area, stresses mainly applied to the miniplates."
Apply the same principle also for the other figure legends.
Figure legends has been changed following your recommendations.
Discussion
- Line 235: "loud -sharing" should be "load-sharing"
"loud -sharing" has been changed to "load-sharing”
- There are only a few studies about this topic in literature (and those cited are by the authors themselves). This limitation, together with the need for more clinical data to support the authors' conclusions, should be mentioned.
This limitation has been added in the discussion:
Finally, there are only a few studies about the biomechanical impact of mandibular titanium miniplates after bone healing in case of facial trauma recurrence in literature, and mainly conducted by our research team. More clinical and experimental data are required to support our conclusion.
I hope we were able to meet your expectations.
Yours faithfully.
Round 2
Reviewer 2 Report
Some comments are not carefully addressed by the authors. Especially on the methodology part. Instead of explaining it to the reviewers, it is suggested to explain it in the discussion part of the manuscript too.
By right, before doing a design study using FEA, the model (in your case is mandible bone and mini plate model) should undergo mesh convergence or sensitivity analysis. This one is a basic step in FEA. If not doing mesh convergence or sensitivity analysis, how we can know if the mesh size of 1mm that you applied to the model is the correct one?
Since the authors only focus on stress, again it is better to include a stress histogram. Please google what it is. Otherwise, other parameters can be added to the manuscript.
For the new labeling of maximum stress, I can see there are contradictions with the one in the contour plot scale. Below the scale, it indicates max=...... But the labeling is different. Please check and justify.
Minor editing of the English language required
Author Response
Dear Reviewer,
On behalf of my co-authors, I would like to thank you for your valuable comments about our manuscript.
Please find below our answers to your comments:
By right, before doing a design study using FEA, the model (in your case is mandible bone and mini plate model) should undergo mesh convergence or sensitivity analysis. This one is a basic step in FEA. If not doing mesh convergence or sensitivity analysis, how we can know if the mesh size of 1mm that you applied to the model is the correct one?
In our model, the miniplates were 1 mm thick. Then, a mesh with elements larger than 1 mm could not be used, in order to respect the geometry of the miniplates. The geometry of the mandible and facial bones with teeth is complex. Our model already contained 88,515 nodes and 453,535 elements. We remeshed our model using elements with a 0.5 mm length. This model contained 489,976 nodes and 4,840,741 elements. We tried to refine the mesh, but calculations were not possible.
Then, as we cannot use many mesh sizes to make a convergence analysis, we were only able to compare a 1 mm element size versus a 0.5 mm element size, These two sizes are consistent with mandibular FEA in the literature.
So we compared the Von Mises stress distribution between the two models. There were few differences between the models. Figure 3 showing these Von Mises Stress distributions in the two models have been added. We added a sentence in the Materials and methods section and a comment in the discussion section explaining why we did not performed a convergence analysis.
“Since the structure is complex, it was not possible to apply hexamesh; then, the tetrahedral mesh was used with an element of 1 mm. The meshes of the mandibular teeth and their alveolar surface were adapted. To ensure that the size of the mesh elements had no impact on the stress distribution, we compared the mesh used with 1 mm elements with a mesh using 0.5 mm elements.”
“The size of the mesh elements was constrained, in particular by the thinness of the miniplates. The maximal size possible was 1 mm. Thus, it was complicated to carry out a convergence analysis with a sufficient number of meshes with different element sizes without requiring the use of extremely small elements for a model of this size, leading to computational difficulties. Then, we decide to compare our model with a mesh including 1 mm elements to a smaller one with 0.5 mm elements. We did not find any major difference between the stress distribution and the maximal stress value.”
Since the authors only focus on stress, again it is better to include a stress histogram. Please google what it is. Otherwise, other parameters can be added to the manuscript.
We included the stress range histograms. These histograms showed few difference between the models without and with miniplates, except in case of impact on the symphysis, where the histograms showed a shift towards low stress values in the presence of the miniplates. We hypothesized that in this case stresses are absorbed by the miniplate explaining the reduction of the stress in the mandibular bone.
In the other histograms, stress level distribution was similar, but the spatial distribution was different. In this case we hypothesized that stresses pass through the miniplates, and then are transmitted to the bone around the miniplates.
We included the figure 8, and add comments in the results and in the discussion
“Range stress histograms (figure 8) showed that stress level distribution was rather similar between the models without and with miniplates. Only in case of impact on the symphysis, at a distance from the miniplates, the range stress histogram showed a shift towards low stress values in the presence of the miniplates….”
“In fact, in case of impact at a distance from the miniplates, stresses are absorbed by the miniplate, more rigid, and weakly transmitted to the bone. It could explain the shift towards low stress values in the stress range histograms of the bone in case of impact at a distance from the miniplates. In case of impact on the miniplates, the stress histogram showed no difference in term of stress level distribution. However, the Von mises distribution was different depending on the presence of miniplates, with less stresses on the bone under the miniplates and more stresses on the bone at the border at the miniplates. In this case of direct impact on the miniplates stresses were transmitted to the bone close to the edges of the miniplates.
For the new labeling of maximum stress, I can see there are contradictions with the one in the contour plot scale. Below the scale, it indicates max=...... But the labeling is different. Please check and justify
In some cases, the maximal stress values were near the condyle. As the temporomandibular joint’s anatomy was simplified and the boundary-conditions imposed to the condyles, we did not focus on the description of the Von Mises stress distribution in the area of the condyles. Then, we found more useful and valuable to look for the max stress value in the area of interest near the impact and the miniplates.
As it was not sufficiently explained, we add a commentary and a justification in each figure caption as follow:
“Max. value: maximal stress value (MPa) in the area of interest, i.e. in the vicinity of the impact and the mini-plates. (The maximal stress values in the condylar region have not been taken into account deliberately, because they are close to the boundary conditions, and then difficult to interpret as they could be influenced by the boundary conditions.)”
I hope we were able to meet your expectations.
Yours faithfully.

Round 3
Reviewer 2 Report
The authors have carefully addressed all concerns from reviewers.
Minor improvement is needed. Better to proofread the manuscript by a native speaker